# Summatory Effects of Anaerobic Exercise and a ‘Westernized Athletic Diet’ on Gut Dysbiosis and Chronic Low-Grade Metabolic Acidosis

**DOI:** 10.3390/microorganisms12061138

**Published:** 2024-06-03

**Authors:** Jesús Álvarez-Herms

**Affiliations:** Phymolab, Physiology and Molecular Laboratory, 40170 Collado Hermoso, Segovia, Spain; phymolab@gmail.com

**Keywords:** acid base, pH, metabolic acidosis, anaerobic exercise, animal protein diet

## Abstract

Anaerobic exercise decreases systemic pH and increases metabolic acidosis in athletes, altering the acid-base homeostasis. In addition, nutritional recommendations advising athletes to intake higher amounts of proteins and simple carbohydrates (including from sport functional supplements) could be detrimental to restoring acid-base balance. Here, this specific nutrition could be classified as an acidic diet and defined as ‘Westernized athletic nutrition’. The maintenance of a chronic physiological state of low-grade metabolic acidosis produces detrimental effects on systemic health, physical performance, and inflammation. Therefore, nutrition must be capable of compensating for systemic acidosis from anaerobic exercise. The healthy gut microbiota can contribute to improving health and physical performance in athletes and, specifically, decrease the systemic acidic load through the conversion of lactate from systemic circulation to short-chain fatty acids in the proximal colon. On the contrary, microbial dysbiosis results in negative consequences for host health and physical performance because it results in a greater accumulation of systemic lactate, hydrogen ions, carbon dioxide, bacterial endotoxins, bioamines, and immunogenic compounds that are transported through the epithelia into the blood circulation. In conclusion, the systemic metabolic acidosis resulting from anaerobic exercise can be aggravated through an acidic diet, promoting chronic, low-grade metabolic acidosis in athletes. The individuality of athletic training and nutrition must take into consideration the acid-base homeostasis to modulate microbiota and adaptive physiological responses.

## 1. Introduction

The control of the systemic and cellular acid-base balance is vital to maintain physiological homeostasis in humans. The narrow range of blood pH needs to be maintained between 7.35 and 7.45 (mean pH 7.4) [1]. The reduction of blood pH levels below 7.35 promotes metabolic acidosis, where in non-clinical conditions, both the central nervous system and immune responses activate different innate buffer mechanisms to restore homeostasis (hemoglobin, bicarbonate, phosphates, and plasma proteins such as albumin). Increased systemic acidosis is a physiological condition resulting from the respiratory system (difficulty in removing carbon dioxide from the lungs) and/or metabolic sources (loss of bicarbonate, elevated acid production, and reduced ability of the kidneys to excrete excess acids) [2].

There are several physiological circumstances that can lead to the dysregulation of acid-base homeostasis. Athletic exercise, especially those involved in anaerobic/intermittent sports, are more likely to exhibit metabolic acidosis. Exercise intensity increases the levels of cell and systemic lactate and hydrogen ions [3], decreasing the systemic pH, sometimes below the homeostatic levels (<7.3) [4]. In basal conditions, another exogenous stimulus that affects acid-base homeostasis in athletes is diet and hydration. Anaerobic athletes typically adhere to acidic diets, which are characterized by substantial intakes of animal proteins and simple carbohydrates (here defined as ‘Westernized athletic nutrition’; WAN). In this regard, recent studies have even questioned whether animal sources of protein are more beneficial than other vegetable protein sources for muscle mass synthesis and strength [5].

Furthermore, acidic diets increase the risk of gastrointestinal stress and gut microbiota (GM) dysbiosis and elevate the systemic acid load in comparison to other plant-based sources of proteins and complex carbohydrates. [6]. An important fact is that altering different parts of the gut is related to the change of pH during the digestion and metabolization of animal proteins and simple carbohydrates. The change in the pH promotes adaptive bacterial specialization of the GM [7]. The GM is the community of microorganisms that colonize the gut from birth and mature dynamically in parallel with the host throughout life. The interaction between the host and the GM is of particular interest in athletes because it involves different physiological functions such as digestive, metabolic, endocrine, and immune [8,9]. On the other hand, the persistence of stimulus that negatively impairs GM homeostasis surely impairs the local immunity, establishes inflammation itself, and elevates systemic endotoxemia from an intestinal ‘gut leaky syndrome’ condition to the whole body [10]. In this regard, it has been reported that higher systemic endotoxemia aggravates the inflammatory response in the tissues, elevating systemic acidosis, too [11]. Therefore, promoting a healthy GM would be beneficial for athletes because it improves gut functions, reduces local and systemic inflammation, and contributes to modulating metabolic acidosis in athletes [12,13]. Recently, it has been demonstrated how the gut, and more particularly a certain composition of the gut microbiota (GM), can contribute to regulating metabolic acidosis by recycling lactate into short-chain fatty acids (SCFAs), mainly in the colon [14]. This process takes place through bacterial communities, which in turn modulate the gut pH in a healthy way [15].

Hence, the main objective of this narrative review is to show how great specificity of exercise (anaerobic/intermittent sports) in athletes can promote a chronic state of systemic acidosis when, moreover, it is accompanied by elevated intake of acidic macronutrients such as animal proteins and simple carbohydrates. The maintenance of a chronic state of systemic acidosis has negative consequences on health and physical capacities. Furthermore, the impairment of the GM composition, promoting bacterial dysbiosis, can reduce physical performance in athletes through multiple functions related to metabolism, immunomodulation, endocrine, and musculoskeletal and/or neural activity. The main objectives of this review were the following: (i) to describe how high-intensity exercise promotes acid-base disturbance, (ii) how specific nutritional recommendations in athletes (high protein, simple carbohydrates, and supplements, here termed as WAN) contribute to aggravating systemic acid load, and (iii) describe the importance of GM balance to co-ordinately modulate acid-base balance through the bacterial potential to buffer acidosis through the gut.

## 2. Methods

This review article was prepared using a narrative approach, contextualizing the topic as to how the metabolic acidosis derived from a physical and nutritional regimen in anaerobic athletes may promote a chronic low grade of acid-base disbalance. Moreover, a dysbiosis of the GM surely appears in athletes following specific acid diets, which are advised. Different databases, including PubMed, Medline, Google Scholar, and Scopus, were used to search for articles for this article, last accessed on 10 April 2024. Common keywords used to search for articles were the following: “effects of diet on metabolic acidosis and exercise”, “metabolic acidosis and systemic inflammation”, “high-intensity exercise, pH and acid base homeostasis”, “anaerobic exercise effects on gut microbiota”, and “high animal protein diet, gut microbiota and metabolic acidosis”. Articles were chosen for inclusion based on the information they described and were incorporated throughout this paper.

## 3. Effects of Anaerobic Exercise on Acid-Base Homeostasis

The anaerobic metabolism implied during high-intensity exercise involves the activation of phosphocreatine and the glycolysis pathways, with a limitation on time [16]. Exercise intensity increases muscle and systemic acidosis, which leads to a decrease in systemic pH through the accumulation of lactate and hydrogen ions [16]. In these sports disciplines, training goals include (i) improving the efficiency of anaerobic pathways to produce energy and (ii) higher recycling and tolerance for lactate and acidosis during and after exercise [16].

Anaerobic metabolism increases the excretion of acids through urine and carbon dioxide during respiration to maintain acid-base homeostasis [17,18]. Therefore, reducing the systemic pH through exercise intensity encompasses lowering acid-base and plasma bicarbonate levels, which, when elevated, can result in a decrease in glomerular filtration by enhancing proximal bicarbonate reabsorption and enhancing levels of angiotensin and mineralocorticoids [1,19]. According to Lindinger and Robergs et al. [17,20], any form of physical activity, even submaximal, leads to an increase in acid production and stress on the body’s buffer systems. In events of high anaerobic intensity, such as a 400 m sprint at maximum intensity, blood pH levels are reduced to levels of 6.8–6.9 [18,21]. Elevation of cellular and systemic acidosis leads to a more perceived fatigue [22] and reduced rate of muscle contraction and anabolism of muscle proteins [23]. 

In summary, anaerobic sports cause a chronic stimulus over the acid-base homeostasis, which can impair adaptive physiological responses in different ways if maintained. The musculoskeletal adaptation can be impaired by [24] (1) negative protein degradation, increasing muscle breakdown [25,26,27,28], (2) impaired mitochondrial function and reduction of energy production, and (3) lower muscle contraction (force) [29]. Moreover, systemic acidosis also impairs tissue oxygenation due to the Bohr effect [30], bone demineralization, increased fracture risk [26], and joint structures degenerate faster [31] (see Figure 1).

## 4. Effects of Nutrition on Metabolic Acidosis in High-Intensity Sports

The actual model of sports nutrition advocates regular intakes of simple carbohydrates, nutritional supplements, and proteins before and after exercise (here defined as ‘Westernized athletic nutrition’) to restore burning energy and structural protein synthesis [32]. Here, we would like to present an alternative viewpoint based on the premise that systemic health is before acute performance to stimulate the innate biological potential (see Figure 1 and Figure 2). Nowadays, human metabolism and energy homeostasis are studied from a mechanistic perspective—that is, “fuel is burned, energy needs to be restored”—this established concept is widely accepted [32] because it does not characterize human adaptive responses as a living structure where epigenetic modifications exist in a narrow range. In fact, during the last decade, the consumption of nutritional supplements such as fast-absorption carbohydrates or whey proteins has grown exponentially with a reductionist character [32]. The healthiness regarding the chronic consume of these products in athletes requires proof, and longitudinal investigations to support their global commercialization. 

The primary feature of diet-induced metabolic acidosis is the shift over decades or even centuries from an ancestral alkaline diet, high in fruits and vegetables, to a ‘Westernized’ diet composed of foods derived from animals (proteins) and deemed ‘acidogenic’ [28,33,34,35,36,37,38,39]. This kind of consumption decreases the consumption of other foods high in bicarbonate and potassium [35], as well as the deficiency of other minerals that form bases, like calcium and magnesium [37], which are generally present in fruits and vegetables [28,33,34,35,36,37,38,39]. This feature has contributed to an increase in diet-induced systemic acidosis, along with a higher consumption of simple sugars [40]. A higher need for pH-buffering homeostatic mechanisms is encouraged by the chronicity of acidic diets in comparison to basic foods [41]. Dietary inflammatory processes are activated in a state of systemic acidosis [42], where a higher intake of animal proteins and simple sugars raises levels of pro-inflammatory markers such as C-reactive protein (CRP), IL-6, and fibrinogen [42,43,44] in comparison with diets of vegetables, fruits, and healthy fats from olive oil or oily fish [42,45].

In relation to simple carbohydrates, many scientists recommend eating more simple carbs before, during, and after exercise; however, these practices can inhibit predictive adaptive responses related to innate metabolic processes (see Figure 2). In this regard, constantly replenishing energy fuel may reduce the innate biological potential because it minimizes better metabolic efficiency and systemic buffer capacity (see Figure 2). Previous studies put in doubt the real efficacy of ‘Westernized Athletic Nutrition’. Maughan and Williams [46] found that muscle glycogen levels did not change after fasting for 24 h to 82 h when exercising at a low level. To Piehl [47], at 90% of the maximum oxygen uptake intensity, the pattern of glycogen depletion of muscle fiber types during cycle ergometer exercise was not affected by the initial glycogen content of the muscle. Jansson and Kaijser [48] found that eating more fat and less carbs 5 days before exercising at 65% of maximum oxygen uptake did not significantly reduce muscle glycogen, increase the amount of fat used in oxidative metabolism, or decrease CHO use during exercise. Despite the fact that actual nutritional recommendations advocate for eating more carbs regularly to improve physical performance, other studies have shown an innate predictive response, as cited above, is really more effective if stimulated in the long term [49,50,51,52]. Related to fast carbohydrate products, we had reported that elevated consumption of sports functional products might impair GM and host health physiology in the long term [12,53]. Recently, Moreno-Pérez et al. [54] have reported that 10 weeks of protein supplementation promoted changes in the GM with reductions of beneficial bacteria such as *Roseburia*, *Blautia*, and *Bifidobacterium longum.* Consequently, specialized sports nutrition guidelines for anaerobic activities might not be the healthiest way to counteract the metabolic acidosis brought on by exercise.

Animal protein is the biggest source of dietary acid because its metabolism leads to the formation of sulfuric acid and hydrogen ions in the body. The results of supplementing the colon microbiota with more protein showed a significant decrease in the proportions of beneficial bacteria like *Bifidobacteria* spp., *Roseburia*, or *Eurobacterium rectale* and a significant increase in pro-inflammatory bacteria like *Clostridium perfrigens*, *Enterococcus*, *Shigella*, and *Eschericia coli* spp. that produce remarkable levels of ammonia [55]. Therefore, an elevated intake of animal proteins can promote the proteolytic bacteria growth in the colon favoring a context of gut dysbiosis. This gut environment changes the pH of the colon and reduces the formation of SCFAs and lactate [56]. The amount of protein fermentation that occurs in the colon depends on the amount of dietary protein ingested but also on the proportion of protein obtained through non-fermentative pathways. These endogenous sources can come from pathological processes such as those associated with inflammatory and/or ulcerative conditions of the intestine [55].

The increase in metabolic acidosis due to the increase in the intake of animal protein also affects the muscles and connective tissues for the formation of ammonia [57,58]. Therefore, producing large amounts of ammonia to excrete large amounts of acid can have long-term health consequences [58]. The fermentation of protein substrates results in the production of toxic metabolites in the colon, which have the potential to cause harm to the gastrointestinal tract [55]. Although theoretically, the regulation of muscle mass is mediated by a net balance between breakdown and synthesis, the bioavailability of essential amino acids and proteins is also influenced by the actions of GM [6]. Therefore, the stimulus required to promote muscle protein synthesis requires primarily muscle activation with exercise but also nutrient bioavailability [59]. According to Lynch et al. (2018), a higher intake of animal protein as opposed to vegetable protein does not guarantee its beneficial effects on intestinal and systemic health. Furthermore, at the sporting level, physical performance and even strength do not seem to be altered in the long term [5,6]. It is necessary to achieve an equilibrium between the quantitative nutrient load and the GM capacity to digest, metabolize, and absorb.

From a digestive perspective, protein fermentation in the colon can produce harmful putrefactive metabolites such as branched-chain fatty acids (BCFAs), ammonia, amines, hydrogen sulfides, phenols, and indoles [60,61,62]. These metabolites possess bioactive properties and induce signaling and gene expression in the host [63], which are potentially associated with health problems [60] because of changes in the intestinal pH and the GM composition [63]. These adaptive colonic changes can promote pathogenic bacterial growth related to cancer, inflammatory bowel disease [64], and cardiovascular disease [65], among other physiological conditions. In excess, the process of protein fermentation may result in gastrointestinal dysbiosis over a prolonged period, thereby affecting the interconnection between the gastrointestinal tract and the central nervous system (CNS) [66]. For instance, the production of amines and sulfur hydrogen is utilized by *Bacillus*, *Clostridium*, *Enterobacter*, *Escherichia*, *Fusarium*, and *Salmonella* [13]. Therefore, this aspect fosters controversy about protein intake requirements and recommendations, their origin and the health or dysbiosis context of GM [57]. This elevated proteolytic fermentation exerts toxic effects on the thickness of the epithelial barrier, impairing colonic permeability due to the increases of levels of ammonia, phenols, amines, hydrogen sulfide, and reducing SCFAs [55]. The amount of protein fermentation occurring in the colon depends on the amount of dietary protein ingested and on a proportion of protein obtained through non-fermentative pathways. These sources can come from pathological processes, such as those associated with inflammatory or ulcerative conditions of the intestine [55].

Combining an acidic diet (high animal protein and simple carbohydrate proportions) with high-intensive exercise, metabolic acidosis increases and affects acid-base homeostasis. Specific studies exposed in Table 1 show that the acidic load from nutrition influences basal metabolic acidosis and the consequent exercise performance. Elegant studies from Greenhaff et al. [67,68], Caciano et al. [69], and Niekamp et al. [70] suggested that an acid pre-exercise intake could change the acid-base status and reduce the time to exhaustion through the modification of the respiratory exchange ratio and the buffering capacity. In this regard, Limmer et al. [71] showed a time of 400 m in active subjects, and acidic buffering improved in athletes following an alkaline diet in comparison with an acid diet. In agreement, Niekamp et al. [70] showed that eating a mostly alkaline diet could improve physical performance by reducing the production of non-metabolic carbon dioxide and increasing the levels of bicarbonate in the blood. Recently, other studies, including 323 elite athletes [29], pointed out the importance of ensuring an acid-alkaline balance in the body is necessary to reduce acid-balance stress and improve physical performance and recovery. Related to hydration, Chycki et al. [3] showed that alkaline hydration improved anaerobic performance through better balance of the acid-base system. Different reviews on the topic (see Table 1) have pointed out the important effects of diet on metabolic acidosis and exercise performance.

In summary, intestinal dysbiosis can result in damage to the intestinal epithelium of the upper and lower tract (small and large intestine) and can be exacerbated by a high level of specialization and elevated doses of specific macronutrients. This is an instance of ‘gut leaky syndrome’, a condition in which endotoxemia heightens the likelihood of metabolic acidosis following high-intensity exercise and increases systemic inflammation. Lastly, the effects of GM dysbiosis include elevated metabolic acidosis, inflammation, and immunologic activation.

## 5. Gut Microbiota and Systemic Acid-Base Homeostasis in Anaerobic Athletes

Nutrition plays a key role in acid-base balance [1], regulation of chronic inflammation [13], and GM composition [74]. Following very specific diets directly influences the GM composition and host adaptive physiology [61]. Foods have a renal acid-loading potential, with fruits and vegetables being negative and hydrogen-buffering, while protein- and phosphorus-rich foods have positive rates with high hydrogen production [29]. Western diets with a higher intake of protein foods of animal origin (fish, eggs, and meat), as well as those recommended for athletes (high intake of proteins (between 1.6 and 2.8 g per kg of body weight) and carbs (between 6 to 12 g per kg of body weight daily and 60 to 120 g per hour during exercise) [32] increases metabolic acidosis [75]. The WAN can be detrimental to health and physical performance when sustained over time [1,67,76].

At the intestinal level, the pH is different depending on the upper or lower portion, being very acidic at the gastric level and more alkaline in the last portion of the large intestine (colon). The pH of each part of the intestine favors the regulation of different digestive, enzymatic, metabolic, and microbial growth functions [77]. The pH gradient in the body promotes different reactions in tissues and cells, and in the gut, it is a key factor in favoring digestion, bactericidal action, and the proliferation of different bacterial communities. Intestinal pH gradients modulate the selection of GM communities and their metabolisms, as well as their diversity [78]. In the large intestine, where high bacterial fermentation occurs, beneficial host metabolites (SCFAs) are produced. In the colon, the different parts differ in pH, being more acidic in the proximal colon (5.7) and alkaline in the distal colon (6.7). In the proximal colon, fermentation and formation of SCFAs and lactate decrease the pH, while in the distal colon, their absorption leads to the release of bicarbonate, increasing the pH [79]. Fermentation of complex carbohydrates occurs in the proximal colon and limits nutrient availability in the distal colon, which metabolizes proteins and available amino acids with the release of ammonia and urea, increasing the pH. This difference in pH gradients and the lower availability of nutrients, immune molecules, and bile acids increases the development of divergent microbial communities [79]. Therefore, the degradation of proteins in the colon is more active when the pH is neutral or alkaline, and as the colon progresses to more distal regions, the pH level increases, thereby enhancing the efficacy of protein fermentation. Contrary to SCFA, branched-chain fatty acids (BCFAs) originate exclusively from the fermentation of branched amino acids [80]. Protein fermentation yields urea, BCFA, phenols, and indoles that are not produced by human enzymes and are, therefore, distinct metabolites of colonic bacteria [80].

The composition of Western diets can affect blood pH by approximately 0.03 pH units [81,82] and urine (~1.0 pH units [83]. Plant foods and fruits promote systemic alkalinity, while cereals, meats, and cheeses promote acidosis [84]. These changes in intestinal pH are associated with significant changes in *Bacteroideceae*, *Bifidobacterium*, and *Enterobacteriaceae* families and in propionic and butyric acid levels. *Bacteroides* are more sensitive to acidic pH, but *Firmicutes* and *Actinomycetes* are more tolerant to pH 5.5. This environment stimulates butyrate production by bacteria such as *Roseburia*, which is elevated in endurance athletes [85]. An increase in pH produces a greater richness of luminal communities, increasing, for example, the Shannon index of diversity, but conversely, a decrease in pH decreases luminal and mucosal communities. Some luminal communities tolerate higher pH levels, such as the phylum *Firmicutes* (*Lachnospiraceae*, *Ruminococcaceae*, *Erysipelotrichaceae*, *Clostridiceaceae*), and others decrease at lower pH levels such as the phylum *Bacteroidetes* (*Odoribacter*, *Butyricimonas*, *Rikinellaceae*, *Alistipes*, and *Prorphynomonadaceae*, *Parabacteroides*). *Bacteroides* decrease at low pH and decrease levels of *Proteobacteria* (*Citrobacter*, *Shigella*, *Bilophila*, *Fusobacterium*). At the same time, when pH is more acidic, *Veillonellaceae* and *Megasphaera* increase. These two families have been increased in studies that evaluated how performance in maximum intensity endurance exercise was associated with an acidic systemic pH that decreased in athletes with higher levels of these bacteria [14,86,87].

The fermentation of indigestible macronutrients as oligo- and poly-saccharides and proteins takes place in the colon via bacterial for saccharolytic species belonging to the *Bacteroides*, *Bifidobacterium*, *Clostridium*, *Eubacterium*, *Lactobacillus*, and *Ruminococcus* genera and proteolytic and peptidase through species such as *Clostridia*, *Propionibacterium* spp., *Prevotella* spp., *Bifidobacterium* spp., and *Bacteroides* spp. [74,88] Therefore, there exists a specific GM diet response depending on the type of macronutrients ingested in the long term (proteins, dietary fibers, fat) [88]. For example, elevated intake of fiber promotes the growth of protective bacteria such as *Roseburia*, *Blautia*, *Eubacterium rectale*, *Faecalibacterium prausnitzii*, *Bifidobacteria*, *Lactobacilli*, and variations in *Bacteroidetes* proportion depending on the type of dietary fiber [74,89,90,91]. The case of vegetarian and vegan diets have shown that long-term high-carbohydrate fermentation aliments promote GM bacteria dominated by bacteria such as the *Prevotella*, *Clostridium clostridioforme*, *Faecalibacterium prausnitzii*, *Lachnospiraceae*, and *Clostridium ramosum* groups [91]. In contrast, a more acidic diet composed of a high-protein diet promotes a decrease in SCFAs as butyrate and an increase of species with proteolytic activities, such as *Bacteroides* spp. [74,91] Finally, an excess of fats has an indirect impact on the gut microbiota diversity, stimulating the production of bile acids, which, in turn, select the growth species with the ability to metabolize biles acids and/or induce the loss of some species due to the antimicrobial activity of bile acids [74,91].

Finally, the main effect of fermentative nutrients in the colon is the different colonic pH gradients caused by bacterial metabolism. For example, the levels of SCFAs depend mainly on the pH gradient. A lower level reduces acetic acid and propionic acid but increases butyric acid. In contrast, a neutral pH benefits propionic acid. Therefore, regulating and influencing pH homeostatic states through diet may benefit physiological performance and prevent GM dysbiosis states. On the other hand, GM in the colon has the capacity to produce, accumulate, and consume-recycle lactate depending on the different pH gradients that exist. Inhibition by acidification of lactate-consuming species led to their lactate accumulation [78]. Increased production of SCFAs regulates pH favorably for the body and also provides an environment for better absorption of minerals, vitamins, and electrolytes and inhibits the absorption of ammonia and other amines [77,92].

## 6. Influence of Anaerobic Athlete’s Training and Diet on Gut Microbiota

Over the last decade, regular exercise has been shown to affect the composition and functionality of GM, increasing its diversity and being beneficial to health and athletic performance [14,93,94,95,96,97,98,99,100]. However, what do we consider a healthy athletic microbiota? The gut microbiota exerts multiple functions supporting host health. We can assume that healthy gut microbial communities are characterized by a high diversity of taxa, a high richness of microbial genes, and a stable, functional core of the microbiome [101]. In the context of gut health, certain species may support lactic acid recycling, promoting systemic pH balance and may support health and athletic performance [14,102]. Conversely, a reduction of this richness and diversity with a growth of gram-negative lipopolysaccharide (LPS) producing proteobacteria promotes reduced GM versatility and is associated with intestinal dysbiosis [103]. Therefore, dysbiosis promotes epithelial barrier dysfunction and selective permeability, favoring endotoxemia of these LPS into the blood and promoting systemic inflammation. In addition, elevated growth of certain pro-inflammatory and pathogenic bacteria may over-activate the inflammatory and immune response in the gut [103]. This intestinal condition may be aggravated in the context of systemic inflammation [103].

The epithelial barrier of the intestine and its functionality is a key factor in intestinal health because it ensures the regulation and passage of substances between the intestine and the body [101]. Impairment and injury of this barrier (‘gut leaky syndrome’) have many negative implications for systemic health due to a deficit in the passage of beneficial metabolites and nutrients from the gut to the blood and lactate from the blood to the gut to be recycled and provide food for SCFA-producing bacteria [104]. In addition, the intestinal barrier is a dynamic entity that acts as a defense structure of the organism, capable of filtering toxic substances between the intestine and the blood bidirectionally. The chronicity of ‘gut leaky syndrome’ not only worsens health but also reduces high-intensity physical performance [12].

The structures that make up the intestinal barrier are the mucus layer, the cellular epithelium, the membrane proteins (tight junctions) and the micro-organisms in the gut [101]. The mucus covering the intestinal epithelium is a substance composed of more than 98% water and glycoproteins secreted by goblet and epithelial cells [105]. The thickness of the mucus layer is greater in the large intestine than in the small intestine, so in the small intestine, enterocytes, Paneth cells, and immune cells secrete antimicrobial products for host defense [105]. The intestinal epithelium, being waterproof to water and hydrophilic solvents, can absorb only through specific transport routes. Close bonds play a crucial role in the construction of the epithelial barrier and in the regulation of epithelial polarity [101]. The specificity of mucous bacteria is very important for stimulating mucus production and protecting the intestinal epithelium from possible injury and damage from exposure to toxins or pathogenic bacteria. Positive bacteria for the mucosa include species such as *Lachnospiraceae*, *Bifidobacterium bifidum*, *Bifidobacterium longum*, *Ruminococcaceae*, and the *Verrucomicrobia* branch (*Akkermansia muciniphila*). However, in the context of GM dysbiosis, the inner mucous layer of the colon could be colonized by pathogens such as *Acinetobacter* spp. and *Proteobacteria* [56].

It has been demonstrated that the ecosystem formed by the intestinal microbiota has the potential to feed back by altering its environment for its bacterial growth. In addition, the gastrointestinal tract has its own nervous system (SNE) that communicates with the CNS through nerves (vagus), neuromodulators, and neurotransmitters of the sympathetic and parasympathetic branches of the autonomous nervous systems (ANS) (see Figure 3) [8]. The elevation of levels of anaerobic bacteria above positive ranges increases the gut leaky syndrome and systemic inflammation by LPS translocation and other metabolites, increasing metabolic acidosis and loss of health. In fact, lactic acid is also produced during intestinal bacterial fermentation of carbohydrates in anaerobic conditions [104]. Lactate concentrations reached in the intestine can alter the pH and promote positive responses and/or produce harmful effects due to increased acidosis and environmental deregulation [106]. Therefore, elevated intestinal lactate levels can also produce dysbiosis by the growth of pathogenic pro-inflammatory bacteria, such as *Salmonella,* that feed on this acid [10,103]. An elevated systemic acidosis may favor this background as the intestine does not have the ability to adequately regulate its pH.

The GM plays a key role in the maturation of the innate immune system and, in a state of dysbiosis, can affect the ability to cope with systemic infections and combat inflammation. Therefore, alteration of the GM can alter intestinal permeability and increase local inflammation.

According to recent research on the topic, various forms of exercise and nutrition have an impact on GM composition (see Table 2) [14]. Therefore, studies showed that higher gut microbial diversity and taxa bacteria related to greater production of beneficial metabolites such as SCFAs were associated with higher fitness performance [107,108,109,110,111,112]. Bressa and colleagues [93] pointed out two important factors: (1) higher physical activity positively alters the diversity and composition of GM, and (2) the athletic diet is linked to a high-protein diet and low-fiber intake, which could have negative effects on GM and health. According to the amount of protein consumed, other studies have found that rugby athletes and exercise groups have higher microbial diversity [94,113]. Comparing athletes to controls, fecal metabolite production and metabolic pathways are also increased [108], demonstrating that physical activity positively changes the GM composition [112,114]. In more active individuals, the majority of studies found a lower prevalence of *Bacteroidetes* or related genera [93,94,115] and a higher abundance of the phylum *Firmicutes* or correspondent genera [93,94,107,108,109,113,115,116]. *Ruminococcaceae* and *Faecalibacterium*, two genera of *Firmicutes*, have been found to be more prevalent among active participants [94,115,116]. Simultaneously, the genera *Megasphaera* and *Bacteroides* were found to have a lower percentage of living individuals and to be linked to health status [117]. It is interesting to note that among *Verrucomicrobia*, individuals who are active have higher levels of the genus *Akkermansia* [93,94,108]. Numerous studies have demonstrated the importance of this intestinal symbiont for host health [9], as well as its positive effects on the thickness of intestinal mucus, the gut barrier, and immune signaling.

As cited above, the abundance of beneficial bacterial families such as *Lachnospiraceae*, *Paraprevotellaceae*, *Ruminococcaceae*, and *Veillonellaceae* has important benefits for GM homeostasis and exercise performance through lactate and acid recycling [14,93,117,118]. The *Lachnospira* species are known to produce anti-inflammatory short-acid butyrate and respond to high-fiber diets [114]. The case of *Veillonellaceae* is involved in lactate metabolism [102], metabolizing into the short-chain fatty acid (SCFA) acetate and propionate via the methyl-malonyl-CoA pathway [14,102]. The level of athletes and the type and diet regimen promote specific phenotypes of GM. Therefore, it is very important that diet and exercise training be connected to increase specific physiological responses. For example, Fernández-Sanjurjo et al. [111] showed that the final ranking of cyclists on a great bouche was related to specific levels of bacteria such as *Bifidobacterium*, *Coriobacteriaceae*, *Erysipelotrichacea*, and *Sutterellacea.*

Adherence to healthy diets such as the Mediterranean diet has demonstrated elevate alpha diversity and species such as *Paraprevotella* and *Bacteroides* [118].

**Table 2 microorganisms-12-01138-t002:** Effects of high-intensity exercise and specific athletic diets (WAN) on the gut microbiota balance.

Authors	Sample and Type of Study	Results	Effects of WAN and/or Exercise on GM Composition?
Scheiman et al. [14]	Runner athletes (*n* = 15) who ran in the 2015 Boston marathon were compared to a set of sedentary controls (*n* = 10) sequenced on approximately daily samples collected up to one week before and one week after marathon day.	The link between members of the genus *Veillonella* and exercise performance: Increases in *Veillonella* relative abundance in marathon runners postmarathon. Inoculation of this strain into mice significantly increased exhaustive treadmill run time. *Veillonella atypica* improved run time via its metabolic conversion of exercise-induced lactate into propionate.	YES Exercise promotes bacterial specialization and can decrease lactate during exercise.
Bressa et al. [54]	Experimental study Two groups: runners were complemented with a protein supplement (whey isolate and beef hydrolysate) (*n* = 12) or maltodextrin (control) (*n* = 12) for 10 weeks.	Fecal pH, water content, ammonia, and SCFA concentrations did not change, indicating that protein supplementation did not increase the presence of these fermentation-derived metabolites. Increased abundance of the *Bacteroidetes* phylum and decreased the presence of health-related taxa, including *Roseburia*, *Blautia*, and *Bifidobacterium longum*.	YES Protein supplementations affect GM balance and can alter beneficial composition. Long-term protein supplementation may have a negative impact on gut microbiota.
Estaki et al. [95]	Experimental study N = 39 subjects physically fit (22 males and 17 females)	Peak oxygen uptake explained more than 20% of the variation in taxonomic richness after accounting for all other factors, including diet. This higher endurance performance was related to increases in the production of fecal butyrate amongst physically fit participants, identifying increased abundances of key butyrate-producing taxa (*Clostridiales*, *Roseburia*, *Lachnospiraceae*, and *Erysipelotrichaceae*).	YES Cardiorespiratory fitness is correlated with increased microbial diversity in healthy humans, and the associated changes are anchored around a set of functional cores rather than specific taxa. The microbial profiles of fit individuals favor the production of butyrate. Increased microbiota diversity and butyrate production are associated with overall host health.
Allen et al. [107]	N = 32 sedentary subjects Two groups: lean (*n* = 18 [9 female]) and obese (*n* = 14 [11 female]). Six weeks of supervised, endurance-based exercise training (3 d·wk^−1^) that progressed from 30 to 60 min·d^−1^ and from moderate (60% of HR reserve) to vigorous intensity (75% HR reserve). Subsequently, participants returned to a sedentary lifestyle activity for a 6 wk washout period. Fecal samples were collected before and after 6 wk of exercise, as well as after the sedentary washout period, with 3 d dietary controls in place before each collection.	β-diversity analysis revealed that exercise-induce alterations of the gut microbiota. Exercise increased fecal concentrations of short-chain fatty acids in lean, but not obese, participants. Exercise-induced shifts in the metabolic output of the microbiota paralleled changes in bacterial genes and taxa capable of short-chain fatty acid production. Exercise-induced changes in the microbiota were largely reversed once exercise training ceased.	YES Exercise training induces compositional and functional changes in the human gut microbiota but is reversed if a positive stimulus (exercise and/or diet) does not exist.
Fernández-Sanjurjo et al. [111]	A total of 16 professional cyclists competing in La Vuelta 2019 were recruited. Fecal samples were collected at four time points: the day before the first stage (A), after 9 stages (B), after 15 stages (C), and on the last stage (D).	*Bifidobacteriaceae*, *Coriobacteriaceae*, *Erysipelotrichaceae*, and *Sutterellaceae* dynamics showed a strong final performance predictive value (r = 0.83, ranking, and r = 0.81, accumulated time). Positive correlations were observed between *Coriobacteriaceae* with acetate (r = 0.530) and isovalerate (r = 0.664) and between *Bifidobacteriaceae* with isobutyrate (r = 0.682). No relationship was observed between SCFAs and performance. The abundance of *Erysipelotrichaceae* at the beginning of La Vuelta was directly related to the previous intake of complex-carbohydrate-rich foods (r = 0.956), while during the competition, the abundance of *Bifidobacteriaceae* was negatively affected by the intake of simple carbohydrates from supplements (r = −0.650).	YES An ecological perspective more realistically represents the relationship between gut microbiota composition and performance compared with single-taxon approaches. The composition and periodization of diet and supplementation during a grand tour, particularly carbohydrates, could be designed to modulate gut microbiota composition to allow better performance.
Barton et al. [108]	Metabolic phenotyping and functional metagenomic analysis of the gut microbiome of professional international rugby union players (*n* = 40) and controls (*n* = 46) were carried out, and the results were correlated with lifestyle parameters and clinical measurements (e.g., dietary habit and serum creatine kinase, respectively).	Athletes had relative increases in pathways associated with enhanced muscle turnover (fitness) and overall health when compared with control groups.	YES Differences in fecal microbiota between athletes and sedentary controls were associated with exercise and diet regimens.
Cronin et al. [113]	N = 90 healthy Irish male and female Caucasian volunteers. Age between 18 to 40 years and with a body mass index (BMI) of between 22 and 35 kg/m^2^ (predominantly overweight or obese). Two randomized groups were recruited to an exercise-only group (E group) and an exercise plus daily whey protein supplementation group (EP group). A separate parallel group consuming whey protein supplementation but not participating in exercise programs (*p* group) was included in the study as a control. All participants were observed and measured for 8 weeks (*n* = 30 for each group). The exercise-only group (E) participated in an 8-week mixed aerobic and resistance exercise training program. The exercise plus whey protein supplementation group (EP) followed the same exercise program in addition to consuming the once-daily whey protein supplement.	Significant changes in the diversity of the gut virome were evident in participants receiving daily whey protein supplementation. Improved body composition with exercise is not dependent on major changes in the diversity of microbial populations in the gut. The diverse microbial characteristics previously observed in long-term habitual athletes may be a later response to exercise and fitness improvement.	YES Increasing the fitness levels of physically inactive humans leads to modest but detectable changes in gut microbiota characteristics. Regular whey protein intake leads to significant alterations to the composition of the gut virome.
Jang et al. [115]	Bodybuilders (*n* = 15), elite distance runners (*n* = 15), and healthy men in their twenties without regular exercise habits (*n* = 15). All participants were males. 3-day food diary (2 weekdays and 1 weekend day) that reflected habitual dietary intake.	Exercise type was associated with athlete diet patterns (bodybuilders: high-protein, high-fat, low-carbohydrate, and low dietary fiber diet; distance runners: low-carbohydrate and low dietary fiber diet). However, athlete type did not differ regarding gut microbiota alpha and beta diversity but was significantly associated with the relative abundance of gut microbiota at the genus and species level. * Faecalibacterium * , *Sutterella*, *Clostridium*, *Haemophilus*, and *Eisenbergiella* were the highest (*p* < 0.05) in bodybuilders, while *Bifidobacterium* and *Parasutterella* were the lowest (*p* < 0.05). At the species level, intestinal beneficial bacteria widely used as probiotics (*Bifidobacterium adolescentis* group, *Bifidobacterium longum* group, *Lactobacillus sakei* group) and those producing short-chain fatty acids (*Blautia wexlerae*, *Eubacterium hallii*) were the lowest in bodybuilders and the highest in controls. In addition, aerobic or resistance exercise training with an unbalanced intake of macronutrients and low intake of dietary fiber led to a similar diversity of gut microbiota. Specifically, daily protein intake was negatively correlated with operation taxonomic unit and Shannon index in distance runners.	YES High-protein diets may have a negative impact on gut microbiota diversity for athletes.
Han et al. [116]	A team of professional female rowing athletes in China was recruited, and 306 fecal samples were collected from 19 individuals, which were separated into three cohorts: adult elite athletes (AE), youth elite athletes (YE), and youth non-elite athletes (YN).	The microbial diversities of elite athletes were higher than those of youth non-elite athletes. The taxonomical, functional, and phenotypic compositions of AE, YE, and YN were significantly different. Additionally, three enterotypes with clear separation were identified in athlete’s fecal samples, with the majority of elite athletes stratified into enterotype 3, which is strongly associated with athlete performances.	YES Direct association between type of exercise regimen and diet: the versatilities of athlete microbial communities of athletes were found to be associated with dietary factors and physical characteristics of GM profile as a biomarker of physical performance and health.
Vázquez-Cuesta et al. [119]	The study included 60 patients (51.7% females). Classification of subjects into two groups according to the categories of good (1–4) and medium (5–9). Stratification by age group was as follows: children (0–2 years), teenagers (13–18 years), young adults (19–30 years), middle-aged adults (31–48 years), and older adults (49–76 years).	The Mediterranean diet (MD), renowned for its potential health benefits, and the influence of adherence thereto on gut microbiota have become a focus of research. Adherence to MD correlated with alpha diversity, and higher values were recorded in good adherers. Good adherers had a higher abundance of *Paraprevotella* and *Bacteroides* (*p* < 0.001). Alpha diversity correlated inversely with fat intake and positively with non-starch polysaccharides. Evenness correlated inversely with red meat intake and positively with NSPs.	YES Diet has an important influence on GM composition and health, and MD has better prognostic effects on GM than animal protein diets.

## 7. The Chronic Impact of Metabolic Acidosis on Systemic Inflammation: Some Role for the Microbiota?

A complex interaction between acid-base balance and the inflammatory response has been suggested previously in the present article [120]. However, there is evidence that metabolic acidosis increases the innate inflammatory response in cell culture [121] and animals [122]. It is unknown whether a sustained pro-inflammatory state in athletes could impair physical performance and systemic health and promote dysbiosis of the GM.

Whatever inflammatory factors contribute to increased acidosis, especially local inflammation. In athletes, higher exercise intensity plus acidic diets promote frequent clinical and non-clinical inflammatory events, elevating metabolic acidosis. Lactic acid is the byproduct of anaerobic glycolysis but is also produced during the bacterial fermentation of complex carbohydrates under anaerobic conditions in the gut [15]. In recent times, a paradigm shift has occurred regarding the impact of lactic acid on physical performance. It has been suggested that lactic acid may be a favorable metabolite for enhancing other physiological functions, thereby initiating health-promoting processes in other consumer and vessel tissues [15], such as liver or slow muscle fibers [123,124]. In this setting, it is imperative to maintain equilibrium between the lactate-producing and lactate-consuming tissues to prevent an exponential rise in metabolic acidosis and consequently decrease the pH. One such recipient system is the colon, which may be beneficial to the growth of lactic acid-consuming bacteria producing short-chain fatty acids (SCFAs) [14,104]. It is possible that the proportions of bacteria consuming lactate for growth may play an important role in determining plasma lactate concentrations [125] and improving physical performance [14,86]. These species comprise specific species of the phylum *Firmicutes* that possess the capability to generate SCFAs such as butyrate or propionate from lactate [104,126]. Changes in both intestinal and systemic pH can lead to stable changes in metabolite production and species composition in colonic GM [89,127]. In healthy humans, lactate cannot accumulate in the colon and must be recycled in order to maintain an adequate environment and produce SCFAs [15,104,126]. Based on this premise, a healthy GM also facilitates the maintenance of systemic acid-base balance, thereby facilitating lactate recycling and enhancing physical performance [14].

The nature of the inflammatory response to repeated events may promote a loss of effectiveness of the immune system’s response. Therefore, chronic metabolic acidosis has been linked to diseases such as osteoporosis, kidney stones, muscle atrophy, metabolic syndrome, and cancer [128]. Metabolic acidosis increases bone resorption by simply a small reduction in extracellular pH by 0.1 [128]. At the joint level, increased systemic acidosis may deteriorate the prognosis of osteoarthritis by increasing synovial fluid acidosis and aggravating chronic inflammatory arthritis states [39].

The athlete’s adaptive response to training commences with an acute inflammatory process, which, once resolved, constitutes a chronic anti-inflammatory therapy [129]. Chronic training has the potential to safeguard the athlete against acute inflammatory events, enhancing their adaptive capacity and tolerance to other biological stressors. During high-intensity anaerobic or intermittent exercise, tissue hypoxia-reperfusion states can also lead to chronic acidosis, which promotes inflammation at both local and systemic levels, with elevated production of pro-inflammatory cytokines and recruitment of immune cells [2]. The persistent occurrence of hypoxic events at the tissue level in muscle can elevate an inflammatory state and result in sarcoplasmic hypertrophy. The induction of transcriptional responses, such as activation of HIF1 and metabolites of the glycolytic pathway, contributes to the acidification of the microenvironment through positive regulation of lactate production and release by macrophages [130]. Hypoxia and lactic acidosis are known to attract macrophages, resulting in an early pro-inflammatory response with delayed homeostatic reprogramming [2]. In the gut, ischemic events occur during intermittent exercise, promoting higher gut leaky syndrome and aggravating inflammation and metabolic acidosis [17].

## 8. Could the Gut Microbiota Modulate the Systemic Inflammation and Metabolic Acidosis?

Increasing systemic acidosis contributes to the disruption of the intestinal balance, both digestive function and GM status. Melamed and Melamed [131] described how a chronic state of metabolic acidosis had negative effects on pancreatic and biliary functions due to the requirement for greater excretion of bicarbonate in the bile and the deactivation of pancreatic enzymes due to greater acidosis [131]. Acidification of digestive juices can cause pancreatitis, gallstone formation, and reflux [131]. In addition, intestinal dysbiosis could also be derived from infections such as viruses and/or related to an inadequate diet [2,132]. In diarrheal situations, for example, the loss of bicarbonate may decrease the buffering capacity. Therefore, the GM status appears to be an important factor in regulating systemic homeostasis, modulating the systemic inflammation and the immune response through the control of the selective passage of substances from the intestine to the blood, preventing endotoxin leaks that aggravate the inflammatory state and worsening of pathological states [133].

Certain GM species have been associated with anti-inflammatory functions, in particular, *Faecalibacterium prausnitzii* or *Akkermansia muciniphila* [134]. For this reason, the role of diet in modulating GM and, by extension, regulating systemic inflammation is of great interest to athletes. Zheng el al. [135] described how a pro-inflammatory dietary regimen based on foods of animal origin abundantly increased certain pathogenic GM species such as Ruminococcus torques, Eubacterium nodatum, Acidaminococcus intestini, and Clostridium leptum, while Akkermansia muciniphila was enriched in the most anti-inflammatory diet group. Another study with 1425 subjects investigated the relationship between 173 dietary factors and the microbiome response in different pathologies [136]; it was observed how processed and animal foods were consistently associated with higher abundances of Firmicutes, Ruminococcus species of the Blautia genus, and endotoxin synthesis pathways. The opposite was found for plant foods and fish, which were positively associated with short-chain fatty acid-producing commensals and pathways of nutrient metabolism. In summary, diet is a fundamental stimulus to modulate GM and promote or reduce systemic pro- or anti-inflammatory states in athletes through increased production of SCFAs.

## 9. Conclusions and Future Perspective

The recovery from exercise is an important component of the overall sports paradigm because it supports improved performance in the next physical stimulus. During the recovery processes, body adaptive responses involve many physiological and metabolic responses that prepare athletes for the next exercise. The accumulation of chronic stimuli such as exercise and diet is key to promoting adaptive responses where the body allostasis drives in a mode fight or flight state [137,138]. It is important, therefore, to compensate systemic states of stress derived from maximal physiological demands as occurring in athletes during exercise to reduce the overload of systemic thresholds as described above (acid-base, oxygen, thermoregulation, and substrate) (see Figure 1).

Individualized nutrition is one of the best ways of recovering faster and helps host physiology to modulate acid-base homeostasis. In this regard, compensating metabolic acidosis from anaerobic exercise through an alkaline diet and hydration could improve an athlete’s health. Alkaline neutralizers, such as bicarbonate or citrate, naturally regulate acid-base homeostasis within the human body. Acid-base disturbances such as acid-base disturbances impair metabolic processes and other physiological processes, such as oxygen transport or neuromuscular functions. Taking into consideration the timing of athletic preparation and compensating physiological states through diet and hydration would be interesting to advance in the individualization of physical preparation.

Extensive research has been conducted on supporting endogenous acid buffering mechanisms to improve physical performance and recovery. In this regard, the hydric and mineral balance of the body is key to modulating acid-base homeostasis [139,140] and preventing a higher state of metabolic acidosis in athletes [141]. ‘Westernized athletic nutrition’ increases the recommendations to consume higher protein amounts (1.6 to 2.8 g per kg) and carbohydrates after intense exercise, which promote a more acidic systemic state during the recovery phase, decreasing pH buffering capacity and impairing recovery kinetics [140,142]. Therefore, a beneficial strategy for compensating for anaerobic exercise would be an alkalinizing diet to delay the onset of fatigue [143] and accelerate the recovery phase through faster recovery of acid-base balance [144,145]. In addition, we should consider nutrition before and after high-intensity exercise to reduce acid load and promote exercise intensity and recovery. On the other hand, the maintenance of a saccharolytic fermentative microbiota at the colonic level can benefit the athlete of high-intensity activities due to the ability of GM to reduce systemic acidosis and produce metabolites beneficial for health and physical performance (SCFAs).

Trainers, athletes, nutritionists, and physiologists should propose a chronology of training and nutrition individualized for the precise stimuli and take into consideration the variation of the GM ecosystem. For an athlete, training may be the main objective for improving specific performance; however, nutrition and hydration balance are key factors in the regulation of metabolic and systemic function. The addition of stimulus promoting greater systemic acidosis can impair both systemic health and physical performance due to its capacity to increase the inflammatory state and GM dysbiosis. In high-intensity exercise, metabolic acidosis and buffering capacity are a determining factor of performance. Nutrition is a fundamental aspect of balancing the acidic state of exercise with foods with a higher bicarbonate content and a lower amount of animal protein. These types of foods include an ‘alkaline diet’ regime rich in vegetables and fruits and a reduced intake of animal protein and soy (of high biological value) [146]. The study of Naude et al. [146] proposed these recommendations, among others, such as reducing sodium intake or not adding salt because it has a very acidifying effect on the diet. In athletes, carbonated drinks can be very acidic due to their carbonic acid content, which lowers the pH. To compensate for states of systemic acidosis, the use of drinks containing alkaline minerals, including bicarbonate and citrate salts (potassium, calcium), has been widely described.

Different trials with this type of nutritional recommendations, including citrate and bicarbonate salts, have shown improvements in both pain and inflammation in people with arthritis processes [39] and chronic inflammatory pain [147]. Moreover, these studies showed an increase in urine and blood pH [37]. Along these lines, controlling urine pH can help us assess the systemic status of the acid-base [1,28]. This measurement method is easy and economical to assess the effect of dietary and sports interventions and can influence individualization itself [37,39].

In relation to the control of metabolic acidosis, Dinicolantiano and O’Keefe [58] summarized different tests to assess the acid-base status such as the assessment of bicarbonate on an empty stomach or before exercise with ranges between 31–38 mEq/L, blood pH values before exercise of 7.45 to 7.5, levels of ammonia in urine in optimal ranges (15 to 45 μg/dL), or the pH of urine before dinner or after 4 h of ingestion between 6.8 and 7.

Finally, preventing the dysbiosis of the GM must be prioritized in athletes to improve their health and physical performance. During the last few years, it has been reported that a healthy GM improves the physical performance of athletes and prevents injuries and sickness. Related to metabolic acidosis, colonic GM can reduce lactate levels from the circulation during exercise and recovery and contribute to modulating acid-base homeostasis. Therefore, athletes should train on their systemic health, increasing their adaptive thresholds and improving the versatility of the GM to modulate acid-base homeostasis from the gut. ‘Westernized athletic diets’ may contribute to promoting GM dysbiosis with negative consequences in the long-term for physical performance. However, this concept requires further studies to specifically propose precise nutrition for athletes considering their volume and intensity of exercise.

## Figures and Tables

**Figure 1 microorganisms-12-01138-f001:**
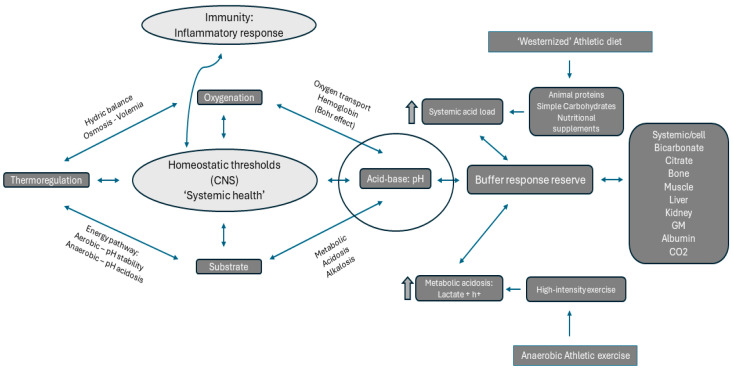
Systemic health describes the efficacy of homeostasis control and interconnection between different physiological thresholds. The body’s homeostatic control modulates coordinatively responses of immune and neural systems (CNS). Acid-base (pH range 7.35–7.45) control depends on systemic/cell buffers. Anaerobic exercise and digestion of some macronutrients (animal proteins and simple carbs) decrease systemic pH activating buffers in a dependent manner. The sum of exercise and diet may induce a double inertial metabolic acidosis that can chronically impair the buffer reserve.

**Figure 2 microorganisms-12-01138-f002:**
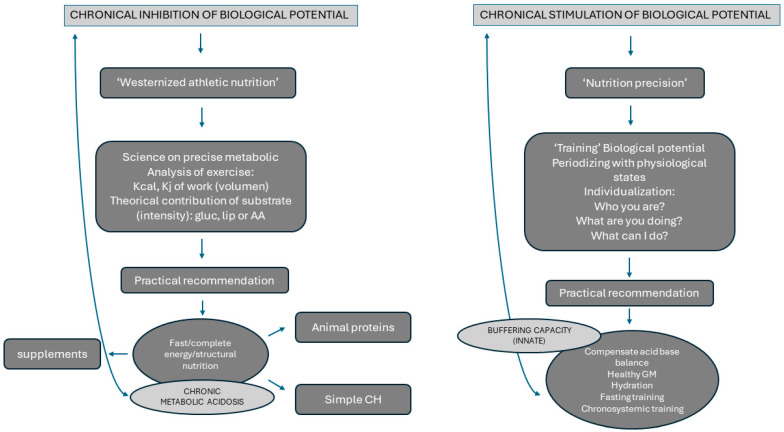
Comparison of biological effects of diet on metabolic acidosis from an acute or chronic stimulation of metabolism.

**Figure 3 microorganisms-12-01138-f003:**
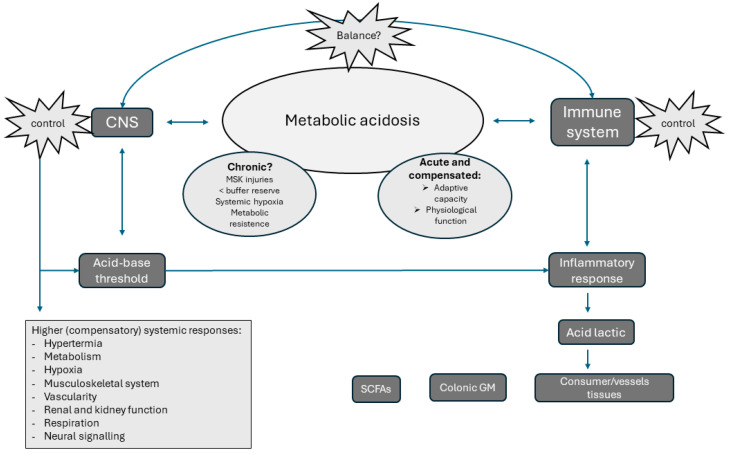
The metabolic acidosis state could be present in acute and/or chronic forms. The homeostatic range is constantly controlled both for central nervous system (CNS) and the immune system. The CNS promotes different systemic (compensatory) responses to regulate acid-base homeostasis through the activation of organs, tissues, and systems. On the other hand, immune system activates the inflammatory response (acute or chronic) to fight against acidosis, present in musculoskeletal (MSK) injuries, cancer, degenerative processes, etc. Acid lactic is a metabolite produced through cell glycolysis and bacterial fermentation in the colon. Afterward, it is consumed and stored in different tissues for multiple functions. The colonic gut microbiota (GM) is an organic vessel turning lactate into short-chain fatty acids (SCFAs), reducing systemic acidosis and positively favoring health. Disbalance between the CNS and immune system (inflammation) can detrimentally impair the compensatory or chronic physiological responses.

**Table 1 microorganisms-12-01138-t001:** Studies describing effects of specific diets on metabolic acidosis and high-intensity exercise performance.

Authors	Type of Study and Sample	Protocol/Intervention	Results
Greenhaff et al. [67]	Experimental design N = 6 Physically active subjects	Four diet protocols for 4 days following a maximal test until exhaustion. Three groups with intervention: (carbs–lipids–proteins) Normal diet (45–41–14%) Low carbs–high fat (3–71–26%) High carbs (73–12–15%) High fat–protein (47–27–26%)	Dietary composition influences acid-base balance by affecting the plasma buffer base and circulating non-volatile weak acids and, by doing so, may influence the time taken to reach exhaustion during high-intensity exercise. High-protein diet elevates metabolic acidosis and alters acid-base balance.
Greenhaff et al. [68]	Randomized study N = 5 Physically active subjects	A total of 3 min at 100% of maximal oxygen uptake on two separate occasions post 4-day diet interventions (cycle ergometer): (a) low carbs, 3%; high fat, 73%; high protein, 24%; or (b) high CHO, 82%; low fat, 8%; and low protein, 10%.	There were no differences between the two treatments in blood acid-base status at rest prior to dietary manipulation. Muscle glycogen content increased by 23% on the (b) diet but was unchanged after the (a) diet. The decline in muscle glycogen content during exercise was 50% greater on the (b) diet. Low-CHO diet could induce metabolic acidosis and may reduce pre-exercise muscle buffering capacity, which may then influence subsequent exercise performance.
Caciano et al. [69]	Cross-over trial randomized and counterbalanced N = 10 Physically active subjects	Graded treadmill test to exhaustion and an anaerobic exercise test on two occasions: after following a low- and high-potential renal load diet (diets were continued as long as needed to achieve an alkaline (4 days) or acid (9 days) fasted morning urine pH state). Anaerobic test until exhaustion lasting 1–4 min.	Maximal exercise Respiratory exhalation ratio (RER) was lower in the alkaline trial compared to the acidic trial (1.10 ± 0.02 vs. 1.20 ± 0.05, *p* = 0.037). The alkaline diet also resulted in a 21% greater time to exhaustion during anaerobic exercise (2.56 ± 0.36 vs. 2.11 ± 0.31 s, *p* = 0.044) and a strong tendency for lower RER values during submaximal exercise at 70% VO_2_max (0.88 ± 0.02 vs. 0.96 ± 0.04, *p* = 0.060). Alkaline-promoting diet resulted in lower RER values during maximal-intensity exercise, and also increased anaerobic exercise time to exhaustion may favor lipid oxidation.
Kim et al. [72]	Experimental study N = 8 Elite Korean bodybuilders	The study investigated the metabolic response to high protein consumption in elite bodybuilders: Diet regimen: protein (4.3 ± 1.2 g/kg body weight/day) and calories (5621.7 ± 1354.7 kcal/day) recorded during three days (breakfast, lunch, dinner, and snacks).	Serum creatinine (1.3 ± 0.1 mg/dL) and potassium (5.9 ± 0.8 mmol/L), and urinary urea nitrogen (24.7 ± 9.5 mg/dL) and creatinine (2.3 ± 0.7 mg/dL) were observed to be higher than the normal reference ranges. Increased urinary excretion of urea nitrogen and creatinine might be due to the high rates of protein metabolism that follow high protein intake and muscle turnover.
Hietavala et al. [19]	Experimental study; randomized N = 88 Three groups: adolescents (12–15 years), young adults (20–35 years), and old subjects (60–75 years) Physically active	A 7-day high-vegetable (alkaline) and a 7-day high-protein diet with no vegetables and fruits in a randomized order. After each diet intervention, incremental cycle ergometer tests were performed until 100% of maximal individual intensity.	In young and old subjects, capillary-pH (*p* ≤ 0.038) and urine-pH (*p* < 0.001) were higher at rest after a high-vegetable diet compared with a high-protein diet. During cycling, capillary-pH was higher (*p* ≤ 0.034) after high vegetable compared with high protein at submaximal workloads in young subjects at 75% of maximal oxygen consumption and older subjects. Older subjects may be more sensitive to the diet-induced acid-base changes.
Niekamp et al. [70]	Experimental study N = 47 sedentary men and women (47–63)	Maximal graded treadmill exercise tests (100% maximal oxygen uptake). Habitual diet was assessed for its long-term effect on systemic acid-base status.	A more alkaline diet promoted higher respiratory exchange ratio values (1.21 ± 0.01, *p* ≤ 0.05) than the middle (1.17 ± 0.01) and highest acidic diet (1.15 ± 0.01). There were no significant differences (all *p* ≥ 0.30) among diets for submaximal exercise intensities of 70%, 80%, or 90% of maximal oxygen consumption. After controlling for age, sex, VO_2_max, and maximal heart rate, regression analysis demonstrated that 19% of the variability in RER was attributed to renal load diets (r = −0.43, *p* = 0.001). Alkaline diets were associated with the attainment of higher peak values for respiratory exchange ratio during maximal-intensity exercise testing.
Chycki et al. [3]	Randomized study N = 16 trained sport athletes Two groups: the experimental group (EG; *n* = 8), which ingested highly alkaline water for three weeks, and the control group (CG; *n* = 8), which received regular table water	Anaerobic performance was evaluated by two double 30 s Wingate tests for lower and upper limbs, with a passive rest interval of 3 min between the bouts of exercise. In addition, acid-base equilibrium and electrolyte status were evaluated. Urine samples were evaluated for specific gravity and pH.	Lactate after the Wingate test was drawn 3 min of recovery and revealed statistically significant decreases in concentration at rest (from 1.99 mmol/L to 1.30 mmol/L with *p* = 0.008) and a significant increase in post-exercise concentration (from 19.09 mmol/L to 21.20 mmol/L with *p* = 0.003) in the experimental group ingesting alkaline water. Additionally, a significant increase in blood pH at rest (from 7.36 to 7.44 with *p* = 0.001), bicarbonate at rest (from 23.87 to 26.76 with *p* = 0.001), and post-exercise (from 12.90 to 13.88 with *p* = 0.002) were observed in the experimental group. The results indicated that drinking alkalized water enhances hydration and improves acid-base balance and anaerobic exercise performance.
Baranauskas [29]	N = 323 competitive Lithuanian high-performance athletes	The actual diet was investigated using the 24 h recall dietary survey method. The potential renal acid load of the diets and net endogenous acid production of athletes were calculated.	A total of 10.2% of athletes exceed endogenous acid production of 100 mEq · day^−1^, and on average 126.1 ± 32.7 mEq · day^−1^ is associated with lower muscle mass (β −1.2% of body weight, * p * < 0.001) but has no effect on the amount of minerals in the body (β 0.01% of body weight, * p * = 0.073). Overall, 25–30% of Lithuanian high-performance athletes use high-protein diets (2.0–4.8 g · kg^−1^ · day^−1^), leading to a dietary acid-base imbalance as well as an excessive production of endogenous acids in the body.
Ball et al. [73]	Experimental study N = 6 Males cycled to exhaustion at a workload equivalent to 95 percent of maximum oxygen uptake on four separate occasions.	Exercise tests were performed after an overnight fast, and each test was preceded by one of four experimental conditions. Two experimental diets were designed, either to replicate each subject’s own normal diet [mean (SD) daily energy intake (E) = 14.5 (0.8), percent protein (Pro), 37.5 (2.2) percent fat (Fat), and 47.5 (2.1) percent carbohydrate (CHO)], or a low-carbohydrate diet [33.6 (1.3) percent Pro, 64.4 (1.5) percent fat, and 2.2 (0.4) percent CHO]. These diets were prepared and consumed within the department over a 3-day period.	Exercise time following the low-CHO diet was less than on the normal diet conditions (*p* < 0.05). Post-exercise blood pH bicarbonate was higher following the ingestion of sodium bicarbonate irrespective of the pre-exercise diet (*p* < 0.05). Blood lactate concentration was higher 2 min after exercise following the N diet with sodium bicarbonate when compared with the low-CHO diets with either sodium bicarbonate or placebo (*p* < 0.05). Plasma ammonia accumulation was not significantly different between experimental conditions. Low-CHO diet reduces the capacity to perform high-intensity exercise, but it appears that the metabolic acidosis induced by the low-CHO diet is not the cause of the reduced exercise capacity observed during high-intensity exercise under these conditions.
Limmer et al. [71]	Experimental study N = 11 Recreationally active participants (8 men, 3 women)	One trial under each individual’s unmodified diet and subsequently two trials following either 4 days of an alkalizing (BASE) or acidizing (ACID) diet. Trials consisted of 400 m runs at intervals of 1 week on a tartan track in a randomized order.	A 400 m performance time for the BASE trial (65.8 ± 7.2 s) compared with the ACID trial (67.3 ± 7.1 s; *p* = 0.026). BASE diet blood lactate (BASE: 16.3 ± 2.7; ACID: 14.4 ± 2.1 mmol/L; *p* = 0.32) and urinary pH (BASE: 7.0 ± 0.7; ACID: 5.5 ± 0.7; *p* = 0.001) were different. A short-term alkalizing diet may improve 400 m performance time in moderately trained participants. Higher blood lactate concentrations under the alkalizing diet suggest an enhanced blood or muscle buffer capacity.

## Data Availability

Not applicable.

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
