# Peer review of "Summatory Effects of Anaerobic Exercise and a ‘Westernized Athletic Diet’ on Gut Dysbiosis and Chronic Low-Grade Metabolic Acidosis"

_microorganisms, 2024, doi:10.3390/microorganisms12061138_

Round 1

Reviewer 1 Report

Comments and Suggestions for Authors

An interesting review paper. The author has well-organised the content. The manuscript needs some revision to enhance its quality:

(I wished there were number lines so it becomes easier to give comments)

1) Abstract - the abstract seems like a summary of the topic, rather than a summary of the review article. I would suggest clearly mentioning the aim of the literature review in the abstract.

2) Introduction - "Furthermore, we can hypothesize that improving GM composition through optimal compensatory diet can improve physical performance in athletes and reduce the chronic low grade of metabolic acidosis reached with exercise and acidic nutrients" - We do not add a hypothesis to a narrative review as this is not an experimental work. This statement needs to be reworded or changed.

"To date, no studies have reported the possible effect of a chronic low grade of metabolic acidosis on the GM status of anaerobic athletes." - This statement doesn't sound suitable. It would have been suitable if this was a research/clinical trial evaluating the effect of a chronic low grade of metabolic acidosis on the GM status of anaerobic athletes. Please delete. 

3) Body - I would suggest adding comprehensive tables including study design, population, participant characteristics, investigation, and main outcomes of clinical trials that have investigated:

a)  effect of diet on metabolic acidosis in high-intensity sports

b) effect of anaerobic athlete’s training and diet on gut microbiota

(two separate tables)

Author Response

Dear Reviewer, thank you for your contribution to improve the present MS. I consider your suggestions as very oportune and concise regarding different topics. 

1) Abstract - the abstract seems like a summary of the topic, rather than a summary of the review article. I would suggest clearly mentioning the aim of the literature review in the abstract.

Done! The abstract has been completely reformulate. It is difficult, however, resume the review in only 200 words.

2) Introduction - "Furthermore, we can hypothesize that improving GM composition through optimal compensatory diet can improve physical performance in athletes and reduce the chronic low grade of metabolic acidosis reached with exercise and acidic nutrients" - We do not add a hypothesis to a narrative review as this is not an experimental work. This statement needs to be reworded or changed.

Thanks, this phrase has been changed.

"To date, no studies have reported the possible effect of a chronic low grade of metabolic acidosis on the GM status of anaerobic athletes." - This statement doesn't sound suitable. It would have been suitable if this was a research/clinical trial evaluating the effect of a chronic low grade of metabolic acidosis on the GM status of anaerobic athletes. Please delete.

Done. Thanks 

3) Body - I would suggest adding comprehensive tables including study design, population, participant characteristics, investigation, and main outcomes of clinical trials that have investigated:

a)  effect of diet on metabolic acidosis in high-intensity sports

b) effect of anaerobic athlete’s training and diet on gut microbiota

(two separate tables)

You are right! There are a lot of studies in the topic during the recent years. I have included in tables the most relevant studies describing diet and exercise effects on GM. Moreover, extense literature has been included in Table 1 related to diet and metabolic acidosis in high-intensive exercise.

If you consider more optimal include other type of studies could be included with other criteria.

Reviewer 2 Report

Comments and Suggestions for Authors

Reviewer’s Comments

Effects of anaerobic exercise and a ‘Westernized athletic diet’ on gut dysbiosis and chronic low-grade of metabolic acidosis consequences by et al., is an important and relevant work of literature that I encourage its publication if the manuscript can be taken through rigorous English language editing. The manuscript describes major works done in metabolic acidosis and exercise: They revealed that anaerobic exercise leads to metabolic acidosis by increasing lactate and hydrogen ions, which lowers blood pH levels. They also indicated that 'Westernized athletic diet,' high in proteins and simple carbohydrates, can exacerbate metabolic acidosis and affect gut microbiota (GM) balance. They, moreover, indicate that the Gut Microbiota's play major roles in human health. That a healthy GM can help reduce systemic acid load by converting lactate to short-chain fatty acids in the colon, whereas gut dysbiosis can lead to health issues and increased systemic acidosis. They expressed a buffering mechanism whereby the body's innate buffering mechanisms, such as hemoglobin and bicarbonate, work to restore acid-base homeostasis affected by exercise and diet.

Major Modifications

Abstract

There is no link between Westernized athletic diet and acidosis in the abstract. Revise to link these two ideas to enhance flow.

The abstract seems disjointed, and ideas seem not to flow smoothly from one point to another. I emphasize authors to revise the abstract to bring out their ideas in the best possible way.

“inflammatory local/systemic responses” should be rephrased.

Introduction

“The acid base balance is a vital physiological threshold in humans.” This sentence requires revision to make scientific sense.

I advise that a table summarizing information regarding “Effects of nutrition on metabolic acidosis in high-intensity sports” will greatly improve the manuscript.

While the information present is needful, factual, and present, the language and English in general has greatly impacted the manuscript, an extensive English editing is needed throughout the entire manuscript.

Comments on the Quality of English Language

Extensive English Editing is needed. 

Author Response

Thank you for your revision and contributions to improve the present MS.

Accordingly to your suggestion I have changed profoundly the MS

Abstract

The abstract has been rephrased completely

Introduction

“The acid base balance is a vital physiological threshold in humans.” This sentence requires revision to make scientific sense.

You are right! This phrase has been revised and make scientific sense.

I advise that a table summarizing information regarding “Effects of nutrition on metabolic acidosis in high-intensity sports” will greatly improve the manuscript.

Thanks! I have included two tables related to main studies in the topic. I have work in the studies regarding effects of exercise and diet on GM (Table 2) and also as diet can promote metabolic acidosis in athletes following intensive regimen of training (table 1).

While the information present is needful, factual, and present, the language and English in general has greatly impacted the manuscript, an extensive English editing is needed throughout the entire manuscript.

Thanks

I have improved english language in the entire MS and reduce tittle

Round 2

Reviewer 1 Report

Comments and Suggestions for Authors

The author has done a great job in improving the manuscript and its content. Well done. Some additional modifications are required for the new tables which have been added. 

In the tables, I would suggest removing the review studies - because they are not original research. Only include original research papers/clinical trials in your tables. 

In addition, the tables need further organisation. Please break down the information into columns, starting with study type (e.g. randomised controlled trial, etc..), then population (sample size, characteristics, etc...), intervention and measurements (what was done and what was measured), and findings (main outcomes). Keep the tables on a landscape sheet.

Author Response

The author has done a great job in improving the manuscript and its content. Well done. Some additional modifications are required for the new tables which have been added.

Thank you very much. Your contribution to improve the MS has been very important.  

In the tables, I would suggest removing the review studies - because they are not original research. Only include original research papers/clinical trials in your tables. 

In addition, the tables need further organisation. Please break down the information into columns, starting with study type (e.g. randomised controlled trial, etc..), then population (sample size, characteristics, etc...), intervention and measurements (what was done and what was measured), and findings (main outcomes). Keep the tables on a landscape sheet.

Thank you!

I have actualized accordingly your suggestions. I consider that improve substantially the MS. Moreover, I have included some text in the section related to GM and specific exercise studies to connect to table (mainly table 2).